

# High nitrogen inhibits biomass and saponins accumulation in a medicinal plant *Panax notoginseng*

Zhu Cun[1,2,3], Hong-Min Wu[1,2,3], Jin-Yan Zhang[1,2,3], Sheng-Pu Shuang[1,2,3], Jie Hong[1,2,3], Tong-Xin An[1] and Jun-Wen Chen[1,2,3]

[1] College of Agronomy & Biotechnology, Yunnan Agricultural University, Kunming, China
[2] National & Local Joint Engineering Research Center on Germplasm Innovation & Utilization of Chinese Medicinal Materials in Southwestern China, Yunnan Agricultural University, Kunming, China
[3] Key Laboratory of Medicinal Plant Biology of Yunnan Province, Yunnan Agricultural University, Kunming, China

Corresponding authors
Tong-Xin An, 1458196769@qq.com
Jun-Wen Chen, cjw31412@hotmail.com

## ABSTRACT

Nitrogen (N) is an important macronutrient and is comprehensively involved in the synthesis of secondary metabolites. However, the interaction between N supply and crop yield and the accumulation of effective constituents in an N-sensitive medicinal plant *Panax notoginseng* (Burkill) F. H. Chen is not completely known. Morphological traits, N use and allocation, photosynthetic capacity and saponins accumulation were evaluated in two- and three-year-old *P. notoginseng* grown under different N regimes. The number and length of fibrous root, total root length and root volume were reduced with the increase of N supply. The accumulation of leaf and stem biomass (above-ground) were enhanced with increasing N supply, and LN-grown plants had the lowest root biomass. Above-ground biomass was closely correlated with N content, and the relationship between root biomass and N content was negatives in *P. notoginseng* ($r = -0.92$). N use efficiency-related parameters, NUE (N use efficiency, *etc.*), $N_C$ (N content in carboxylation system component) and $P_n$ (the net photosynthetic rate) were reduced in HN-grown *P. notoginseng*. SLN (specific leaf N), Chl (chlorophyll), $N_L$ (N content in light capture component) increased with an increase in N application. Interestingly, root biomass was positively correlated with NUE, yield and $P_n$. Above-ground biomass was close negatively correlated with photosynthetic N use efficiency (PNUE). Saponins content was positively correlated with NUE and $P_n$. Additionally, HN improved the root yield of per plant compared with LN, but reduced the accumulation of saponins, and the lowest yield of saponins per unit area (35.71 kg·hm$^{-2}$) was recorded in HN-grown plants. HN-grown medicinal plants could inhibit the accumulation of root biomass by reducing N use and photosynthetic capacity, and HN-induced decrease in the accumulation of saponins (C-containing metabolites) might be closely related to the decline in N efficiency and photosynthetic capacity. Overall, N excess reduces the yield of root and C-containing secondary metabolites (active ingredient) in N-sensitive medicinal species such as *P. notoginseng*.

## INTRODUCTION

Nitrogen (N) is a determinant nutrient for plant biomass or crop yield (*Khan et al., 2020*). Yellow leaf, dwarfed plant height, low biomass has been observed in N-deficient plants (*Mu & Chen, 2021*). Dark green leaves, immature stems, and the imbalance of vegetative and reproductive growth was observed when plants were exposed to excessive N condition (*Eichelmann et al., 2005*; *Yuan et al., 2005*). An inhibition on root elongation, N uptake, yield and N use efficiency has been recorded in N-excess plants (*Xu, Fan & Miller, 2012*; *Hartman & Tringe, 2019*). Nevertheless, plant biomass and crop yield under different N levels are mainly determined by N uptake and N use efficiency (*Cambui et al., 2011*). Therefore, moderate N supply has a vital significance for improving crop growth and yield.

N supply is closely related to leaf photosynthetic capacity, and photosynthesis is critical for crop biomass and yield (*Adams et al., 2018*; *Evans & Clarke, 2019*). Photosynthetic capacity is increased with increasing N application rates (*Pasandi Pour, Farahbakhsh & Tohidinejad, 2021*). Non-optimal N application significantly weakens photosynthetic efficiency in accompanying with the reduced productivity (*Evans & Clarke, 2019*). Meanwhile, there is positively correlation between crop yield and photosynthetic N use efficiency (PNUE) (*Hou et al., 2019*; *Lei et al., 2021*). A reduction in PNUE by lowering the proportion of N allocation in photosynthetic organs (*e.g.*, carboxylation and bioenergetics components), results in a lowed yield of N-excess *Brassica campestris* L. (*Liu et al., 2016*). It has commonly accepted that differences in PNUE are mainly caused by differences in photosynthetic efficiency (*Poorter & Evans, 1998*; *Harrison et al., 2009*). Meanwhile, the optimized allocation of leaf N in photosynthetic component might dramatically improve 60% photosynthetic capacity (*Zhu, de Sturler & Long, 2007*; *Onoda et al., 2017*). Hence, more evidences are required to investigate the effects of N excess on photosynthetic efficiency and crop yield.

Nitrogen availability might also alter carbon/nitrogen (C/N) balance and consequently change the content of C- and N-containing secondary metabolites in the medicinal species (*Song et al., 2021*). Excessive N application reduces the concentration of C-containing metabolites, such as anthocyanin and polyphenolic compounds (*Awad & Jager, 2002*; *Jakovljević, Topuzović & Stanković, 2019*). It has been observed that N deficiency lead to a marked shift from N-rich alkaloid nicotine to C-rich phenylpropanoids or to starch (carbohydrate) in *Nicotiana tabacum* L. (*Matt et al., 2001a*, *2001b*; *Fritz et al., 2006*). Correspondingly, the canavanine (N-containing metabolites) is significantly reduced in medicinal species *Sutherlandia frutescens* (L.) R. Br. grown under low N condition (*Colling, Stander & Makunga, 2010*). The N-containing metabolites (alkaloids) concentration of in the medicinal plant *Datura stramonium* L. is significantly increased with an increase in soluble sugar and proline content (primary metabolites) under N-excess condition (*Alinejad et al., 2020*). Unexpectedly, the responses of C-containing metabolites to N availability are not completely evaluated in the N-sensitive medicinal species.

*Panax notoginseng* (Burkill) F. H. Chen (Sanqi in Chineses) is a perennial medicinal plant and a member of the Araliaceae family, which is a typically shade-tolerant and N-sensitive plants (*Chen et al., 2016*; *Ou et al., 2019*). *P. notoginseng* has been cultivated for more than 400 years, and its root has been used as Chinese medicinal materials for thousands of years. The incidence of root rot is increased in *P. notoginseng* grown under N-excess condition (*Xia et al., 2016*; *Zhang, Cun & Chen, 2020*). Meanwhile, leaf biomass and photosynthetic capacity are decreased in two-year-old *P. notoginseng* under excessive N condition (*Zhang, Cun & Chen, 2020*; *Cun et al., 2021*). A significant decrease in root, leaf and stem biomass has also been observed in three-year-old *P. notoginseng* grown under N deficient N condition (*Wei et al., 2015*). Triterpenoid saponins (C-containing metabolites) are the index compounds for the quality of Notoginseng Radix (*Pharmacopoeia of People's Republic of China, 2020*). Low N and high potassium (K) increase the content of saponin through promoting photosynthesis and saponin biosynthesis-related genes expression in *P. notoginseng* (*Ou et al., 2020*). Appropriate N supply (225 kg·hm$^{-2}$) enhances the accumulation of biomass and saponins though optimizing root architecture and N uptake efficiency in *P. notoginseng* (*Wei et al., 2020*). However, more evidences still need to elucidate the relationship between N excess and plant biomass, saponins accumulation in the N-sensitive species *P. notoginseng*.

The present study aimed to shed light on an interaction between N availability and crop yield and saponins accumulation in the medicinal plant *P. notoginseng*. Morphological traits, N use and allocation, photosynthetic capacity, and saponins accumulation were comparatively evaluated in two- and three-year-old *P. notoginseng* grown under low nitrogen (LN), moderate nitrogen (MN) and high nitrogen (HN). We hypothesized that (i) root biomass of *P. notoginseng* might be reduced accompanying with HN-driven inhibition on photosynthetic capacity and NUE (N use efficiency); (ii) HN-driven decrease in saponin accumulation might be reflected by the C/N imbalance; (iii) N stress might reduce the yield of *P. notoginseng*.

## MATERIALS AND METHODS

### Plant materials and growth conditions

The study was conducted at the Yunnan Agricultural University teaching and experimental farm in Kunming, China (102°45′E, 25°08′N), with an average annual rainfall and average annual temperature of about 1,006.7 mm and 14.5 °C, respectively. The properties of raw soil physical and chemical was determined as described by *Long & Sun (2012)*: organic matter content was 3.18%, pH (H$_2$O) was 6.84, total N content was 0.17%, total phosphorus (P) was 0.23%, the available P content was 11.04 mg·kg$^{-1}$, total potassium (K) was 0.24%, and the available K content was 127.32 mg·g$^{-1}$.

A permeable black plastic net was used to create a shade-house for *P. notoginseng*, and the full sunlight irradiance is about 10% (*Chen et al., 2016*; *Zhang et al., 2021*). Meanwhile, LI-1500 photon data collector (LI-COR, Lincoln, NE, USA) was used to determine diurnal variation of photosynthetic active radiation (PAR) for 3 days (Fig. S1). Permeable nets allow full air circulation, minimizing differences in temperature and relative humidity among treatments. In January, Chinese Miao Xiang *P. notoginseng* Industrial Co., Ltd.

(104°32′E, 25°53′N) provided one-year-old *P. notoginseng* seedlings. Subsequently, healthy and uniform seedlings were transplanted into a plastic flowerpot (30 cm × 25 cm × 20 cm) with each containing three rootstocks (Fig. S2). There were 140 pots used for each N levels, totaling 420 pots. Three N regimes (low nitrogen (LN, without N addition), moderate nitrogen (MN, 225 kg·hm$^{-2}$), high nitrogen (HN, 450 kg·hm$^{-2}$)) were designed (*Zhang, Cun & Chen, 2020*; *Zhang et al., 2020*), and each N levels were replicated by seven times. The chemical N, phosphate (P) and potassium (K) fertilizers used were compound fertilizer (32% N, 4% $P_2O_5$), calcium superphosphate (52% $P_2O_5$, 34% $K_2O$) and potassium sulfate (52% $K_2O$), respectively. The same amounts of P (225 kg·$P_2O_5$·hm$^{-2}$) and K (450 kg·$K_2O$·hm$^{-2}$) fertilizers were used in all treatments with the exception of the N fertilizer. Fertilization was applied in four times a year (April, May, July, and August). In each pot, basal doses of P and K at the rates of 0.45 and 0.90 g, respectively (equivalent to 225 and 450 kg·hm$^{-2}$, respectively), were applied at time while N was applied according to the treatments. N fertilizer rate 0 (LN), 0.45 (MN) and 0.90 (HN) g·pot$^{-1}$ (equivalent to 0, 225, and 450 kg·hm$^{-2}$, respectively).

## Plant morphology and biomass allocation

At November, the two- and three-year-old plants were sampled from the experimental farm and then separated into root (main root, fibrous root, and root tuber), stem and leaf in room. The length, width, and area of leaf were measured by LI-3000 leaf-area meter (LI-COR, Lincoln, NE, USA). Root tuber and stem diameter were measured by vernier caliper. Plant height, grown breadth, the length of main root and total root were determined as described by *Zhang, Cun & Chen (2020)*. Root volume was determined by the drainage method.

The samples were dried at 60 °C for 96 h. Dry matter was determined, and these results were used to calculate the percentage of biomass allocation into leaf (leaf mass fraction, LMF), stem (stem mass fraction, SMF), roots (root mass fraction, RMF), as well as root to shoot ratio (RSR). The root yield of per plant and economic yield (root yield of per hectare) were calculated based on root biomass data.

## Determination of chlorophyll content

In 15 mL of acetone-ethanol mixture (2:1 v/v), 0.5 g of fresh *P. notoginseng* leaves were soaked. A standing period of 3 h was followed by a centrifugation of 3,000 *g*·min$^{-1}$ for 10 min. A JASCO V-670 spectrophotometer (JASCO, Hachioji-shi, Tokyo, Japan) was used to measure absorbance at 665 and 649 nm wavelengths. Chl *a*, Chl *b* and Chl *a*/Chl *b* were analyzed as described by *Lichtenthaler (1987)*.

## Measurement of gas exchange parameters

LI-6400XT photosynthesis system (LI-COR, Lincoln, NE, USA) was used to determine photosynthetic gas exchange parameters. Set with a blue light ratio, temperature, photosynthetic photon flux density (PPFD) and $CO_2$ concentration of 10%, 25 °C, 500 μmol·photons·m$^{-2}$·s$^{-1}$ and 400 μmol·$CO_2$·mol$^{-1}$, respectively. Photosynthetic gas exchange parameters (as reflected by net photosynthetic rate, $P_n$) were collected as

previously described in *Cun et al. (2021)* and *Zhang, Cun & Chen (2020)*. Meanwhile, photosynthetic-related parameters were calculated as described by *Webb, Newton & Starr (1974)*, *Xu (2002)* and *Demmig-Adams et al. (1995, 1996)*.

## Calculation of photosynthetic N allocation

The leaf, stem and root N contents were determined by Kjeldagl method (*Bremner, 1960*). Additionally, specific leaf N (SLN) was calculated from the leaf area. Based on the values of $V_{cmax}$ (maximum carboxylation efficiency), $J_{max}$ (maximum electron transfer rate), SLN and Chl contents, $N_C$ (N content in carboxylation system component), $N_B$ (N content in bioenergetics component) and $N_L$ (N content in light-harvesting systems component) were analyzed according to the method described by *Niinemets & Tenhunen (1997)*. N allocation in the photosynthetic system ($N_{photo}$) = $N_B + N_C + N_L$. Photosynthetic N use efficiency (PNUE) = $P_{max}$ (maximum net photosynthetic rate)/SLN.

## Nitrogen use efficiency

Based on the biomass and N contents, N use efficiency (NUE), N agronomic efficiency (NAE), N uptake efficiency (NUPE), recovery of N fertilizer (RNF), N contribution rate (NCR), N partial factor productivity (NPFP) were calculated in *P. notoginseng* grown under different N regimes. The following equations were used to calculate N uptake and use efficiency (*Jamaati-e-Somarin et al., 2008*; *Ning et al., 2012*; *Wu et al., 2016*; *An et al., 2018*; *Gupta et al., 2021*): NUE ($kg·kg^{-1}$) = yield (underground dry weight)/plant N accumulation; NAE ($kg·kg^{-1}$) = (yield with N application – yield without N application)/N rate; NUPE ($kg·kg^{-1}$) = above-ground total N content/N rate; RNF (%) = (above-ground total N content with N application – above-ground total N content without N application)/ N rate × 100; NCR (%) = (yield with N application – yield without N application)/yield with N application × 100; NPFP ($kg·kg^{-1}$) = yield with N application/N rate.

## Saponins content

Dry root samples of 0.3 g were extracted in 100% methanol and sonicated for 30 min. The solution volume was fixed to 25 mL. Saponin contents were determined as described by *Pharmacopoeia of People's Republic of China (2020)*. Saponin contents were measured using a high-performance liquid chromatograph (Agilent 1260; Agilent Technologies, Santa Clara, CA, USA). Notoginsenoside $R_1$, ginsenoside Rd, ginsenoside $Rg_1$, ginsenoside Re, and ginsenoside $Rb_1$ standards ( > 98% purity) were purchased from Yuanye Bio-technology (Shanghai, China). Kit column (250 mm × 4.6 mm, 5 μm) was used for the determination, and the mobile phase was acetonitrile (ACN)-water. Chromatographic conditions: elution with 0–5 min, 17–20% ACN; 5–20 min, 20% ACN; 20–45 min, 20–42% ACN; 45–50 min, 42–100% ACN; set with flow rate, injection volume, monitoring wavelength and column temperature of 1.0 $mL·min^{-1}$, 10 μL, 203 nm and room temperature, respectively. Total saponins are the sum of $Rg_1$, $Rb_1$, Re, Rd and $R_1$. The HPLC chromatograms of *P. notoginseng* root grown in different N environments are shown in Fig. S3.

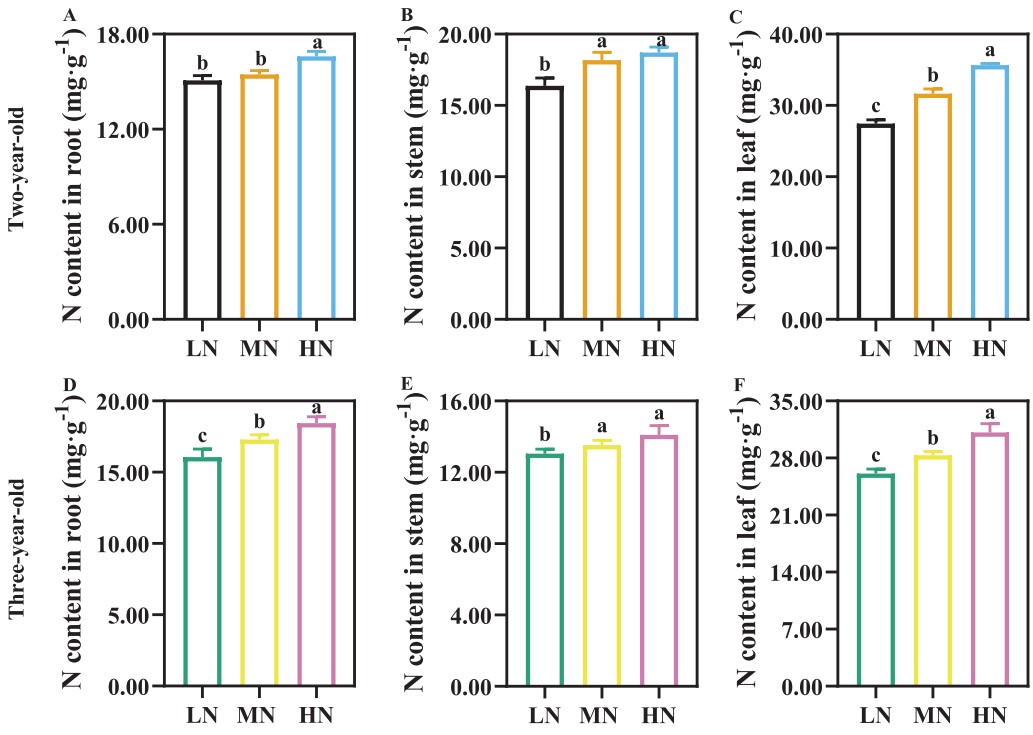

**Figure 1 The nitrogen (N) content of root (A, D), stem (B, E) and leaf (C, F) in *Panax notoginseng* grown under different nitrogen fertilization.** Data are mean ± standard deviation (SD) of seven independent biological replicates performed in septuplicate (*n* = 7). Significant differences are indicated by lowercase letters (one-way ANOVA ; *P* < 0.05).

## Statistical analysis

All data in the tables and figures were mean ± standard deviation (SD) of 5–7 independent biological replicates performed (*n* = 5 or 7). One-way analysis of variance was used to evaluate the effect of N treatment in a year by T-test using SPSS software (IBM SPSS Statistics). LSD-test was used to compare treatment means, with significant effects having *P* < 0.05. Plots were made using Origin 2021 and GraphPad 8.0 software. Pearson correlation coefficients were assessed using Origin 2021. Principal component analysis loading factors were assessed using Origin 2021.

## RESULTS

### Responses of the N content and morphological traits to N regimes

The N content of root, stem, and leaf increased with increasing N supply, and N content in leaf was greater than that in root and stem (Fig. 1). There was not a significant difference in root N content of two-year-old *P. notogisneng* between MN and LN conditions (Fig. 1A, *P* > 0.05). The stem and leaf N content were higher in two-year-old *P. notoginseng* compared with three-year-old plants (Figs. 1B, 1C, 1E, 1F).

Length, width and area of leaf, plants height, stem diameter, and grown breadth were significantly increased with increasing N supply in *P. notoginseng* (Table 1, *P* < 0.05). For two-year-old *P. notoginseng*, main root length and total root length were increased by 31.24% and 11.10% in MN-grown plants compared with HN-grown *P. notoginseng*,

**Table 1 Effect of nitrogen regimes on the morphological traits of *Panax notoginseng*.**

| Variables | Two-year-old | | | Three-year-old | | |
|---|---|---|---|---|---|---|
| | LN | MN | HN | LN | MN | HN |
| Plant height (cm) | 19.27 ± 0.76 b | 22.04 ± 0.68 ab | 24.13 ± 0.71 a | 25.49 ± 1.90 c | 30.69 ± 1.61 a | 28.83 ± 2.19 b |
| Stem diameter (mm) | 3.28 ± 0.07 b | 3.51 ± 0.09 ab | 3.60 ± 0.07 a | 4.39 ± 0.24 b | 4.76 ± 0.21 ab | 5.13 ± 0.09 a |
| Grown breadth (cm) | 25.75 ± 0.76 b | 26.11 ± 0.53 b | 28.94 ± 0.88 a | 22.74 ± 0.87 c | 41.17 ± 0.93 b | 53.58 ± 0.96 a |
| Leaf length (cm) | 3.64 ± 0.30 c | 7.57 ± 0.13 b | 8.84 ± 0.26 a | 6.17 ± 0.29 c | 7.34 ± 0.22 b | 8.27 ± 0.28 a |
| Leaf width (cm) | 2.86 ± 0.084 c | 3.26 ± 0.16 b | 3.64 ± 0.15 a | 2.31 ± 0.07 c | 2.70 ± 0.07 b | 3.16 ± 0.09 a |
| Leaf area (cm$^2$) | 12.11 ± 0.08 c | 15.95 ± 1.29 b | 20.43 ± 1.01 a | 9.73 ± 0.64 c | 13.48 ± 0.58 b | 17.96 ± 0.86 a |
| Main root length (cm) | 12.19 ± 0.91 b | 16.13 ± 1.48 a | 12.29 ± 0.79 b | 13.46 ± 2.29 a | 7.33 ± 1.14 b | 7.65 ± 1.48 b |
| The length of rhizome (cm) | 1.51 ± 0.40 a | 1.17 ± 0.08 a | 1.71 ± 0.23 a | 1.77 ±0.17 a | 2.19 ± 0.23 a | 1.68 ± 0.15 a |
| Root tuber diameter (cm) | 1.25 ± 0.45 a | 1.39 ± 0.61 a | 1.36 ± 0.78 a | 2.73 ± 0.09 a | 2.97 ± 0.05 a | 2.87 ± 0.10 a |
| The number of fibrous roots | 21.1 ± 1.42 a | 20.3 ± 1.70 a | 19.55 ± 2.23 a | 27.00 ± 3.00 a | 15.00 ± 1.00 b | 15.00 ± 1.00 b |
| The length of fibrous root (cm) | 117.43 ± 8.64 a | 118.43 ± 10.75 a | 113.39 ± 11.28 a | 119.31 ± 14.70 a | 73.47 ± 7.17 b | 67.61 ± 6.11 b |
| Volume of root (cm$^3$) | 7.89 ± 0.58 a | 9.25 ± 0.89 a | 9.34 ± 0.76 a | 20.60 ± 2.68 a | 16.54 ± 1.05 ab | 13.62 ± 1.77 b |
| Total root length (cm) | 134.48 ± 11.58 a | 139.64 ± 6.37 a | 125.69 ± 9.47 b | 304.04 ± 21.58 a | 169.53 ± 6.37 b | 132.69 ± 9.47 b |

Note:
Data are mean ± SD of seven independent biological replicates performed in septuplicate ($n$ = 7). Different lowercase letters among nitrogen regimes indicate significant difference (one-way ANOVA, $P < 0.05$).

respectively (Table 1). For three-year-old *P. notoginseng*, these was not a significant difference in root tuber diameter and rhizome length among N regimes (Table 1, $P > 0.05$). The number of fibrous roots, length of fibrous root, total root length, and root volume declined with an increase in N application (Table 1).

**Biomass accumulation and allocation in response to N supply**

Biomass accumulation increased with the increase of cultivation years (Table 2). For two-year-old *P. notoginseng*, main root and total biomass were reduced by 23.56% and 36.42% in LN-grown plants compared with MN-grown plants, respectively (Table 2, $P > 0.05$). Leaf biomass was increased by 145.45% and 125.00% in MN-grown plants compared with LN- and HN-grown *P. notoginseng*, respectively (Table 2). Meanwhile, there was not a significant difference in RSR among N regimes (Fig. 2A, $P > 0.05$). RMF, SMF and LMF were increased by 110.00%, 88.89% and 72.73% in MN-grown plants compared with HN-grown *P. notoginsneng*, respectively (Fig. 2A). For three-year-old *P. notoginseng*, the biomass of the main root, rhizome and fibrous were lowest in HN-grown plants compared with other treatments (Table 2, $P < 0.05$). Leaf biomass was increased by 127.62% and 54.60% in HN-grown plants compared with LN- and MN-grown *P. notoginseng*, respectively (Table 2). RSR and RMF declined with the increase of N application (Fig. 2B, $P > 0.05$). SMF and LMF were improved by 27.59% and 28.57% in HN-grown *P. notoginseng* compared with LN-grown plants (Fig. 2B).

As the time of plant cultivation increased, the yield of *P. notoginseng* increased (Fig. 3). The yield of per plant and economic yield were increased in MN plants (Fig. 3). Root yield of per plant was reduced by 57.27% and 41.07% in two- and three-year-old *P. notoginseng* grown under LN conditions compared with MN treatments, respectively (Figs. 3A, 3C).

**Table 2  Effect of nitrogen regimes on the biomass of *Panax notoginseng*.**

| Variables | Two-year-old | | | Three-year-old | | |
|---|---|---|---|---|---|---|
| | LN | MN | HN | LN | MN | HN |
| Main root biomass (g) | 1.59 ± 0.12 b | 2.08 ± 0.19 a | 1.84 ± 0.10 ab | 4.82 ± 0.43 a | 4.89 ± 0.43 a | 3.87 ± 0.26 b |
| Rhizome biomass (g) | 0.55 ± 0.40 a | 0.64 ± 0.03 a | 0.64 ± 0.04 a | 2.87 ± 0.09 a | 3.01 ± 0.12 a | 1.81 ± 0.30 b |
| Fibrous root biomass (g) | 0.06 ± 0.03 a | 0.74 ± 0.07 a | 0.72 ± 0.05 a | 2.83 ± 0.19 a | 1.88 ± 0.67 a | 1.77 ± 0.42 a |
| Root biomass (g) | 2.20 ± 0.26 b | 3.46 ± 0.15 a | 3.20 ± 0.75 a | 10.52 ± 0.62 b | 9.78 ± 1.65 a | 7.45 ± 1.75 a |
| Stem biomass (g) | 0.10 ± 0.09 b | 0.16 ± 0.03 ab | 0.24 ± 0.06 a | 3.78 ± 0.04 c | 4.61 ± 0.07 b | 8.27 ± 0.27 a |
| Leaf biomass (g) | 0.11 ± 0.01 b | 0.27 ± 0.05 a | 0.12 ± 0.02 b | 3.62 ± 0.04 c | 5.33 ± 0.59 b | 8.24 ± 0.39 a |
| Total biomass (g) | 2.90 ± 0.15 b | 3.89 ± 0.25 a | 3.45 ± 0.18 a | 13.93 ± 0.92 b | 16.23 ± 0.71 b | 22.76 ± 0.81 a |

Note:
Data are mean ± SD of seven independent biological replicates performed in septuplicate ($n = 7$). Different lowercase letters among nitrogen regimes indicate significant difference (one-way ANOVA, $P < 0.05$).

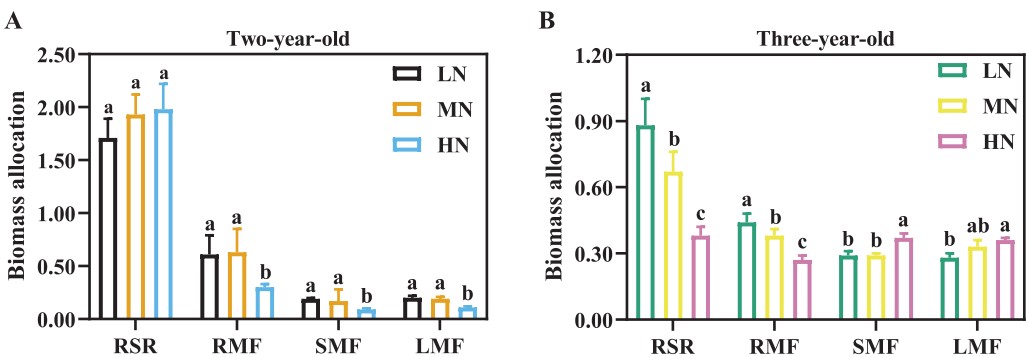

**Figure 2  Biomass allocation in two- (A) and three-year-old (B) *Panax notoginseng* grown under different nitrogen fertilization.** RSR is the root to shoot ratio; RMF is the root mass fraction; SMF is the stem mass fraction; LMF is the leaf mass fraction. Data are mean ± SD of five independent biological replicates performed in quintuplicate ($n = 5$). Significant differences are indicated by lowercase letters (one-way ANOVA; $P < 0.05$).

Economic yield was decreased by 40.51% and 34.76% in two- and three-year-old *P. notoginseng* grown under HN conditions compared with MN treatments, respectively (Figs. 3B, 3D).

## Nitrogen use efficiency in response to N availability

There were considerable differences in N efficiency of *P. notoginseng* under N regimes (Fig. 4, Table 3, $P < 0.05$). NUE was declined by 62.96% and 34.03% in two- and three-year old *P. notoginseng* grown under HN condition compared with MN conditions, respectively (Fig. 4, $P < 0.05$). The minimum values of NAE, NUPE, NCR, and NPFP were obtained in the HN-grown *P. notoginaseng* (Table 3, $P < 0.05$). RNF was increased by 29.57% in three-year-old *P. notoginseng* grown under HN compared with MN condition (Table 3).

## N-driven changes in photosynthetic-related parameters

SLN increased with an increase in N application (Figs. 5A, 5C; $P < 0.05$), and SLN was higher in two-year-old plants compared with three-year-old plants (Figs. 5A, 5C). There was not a significant difference in Chl content of two-year old *P. notoginseng* in LN and

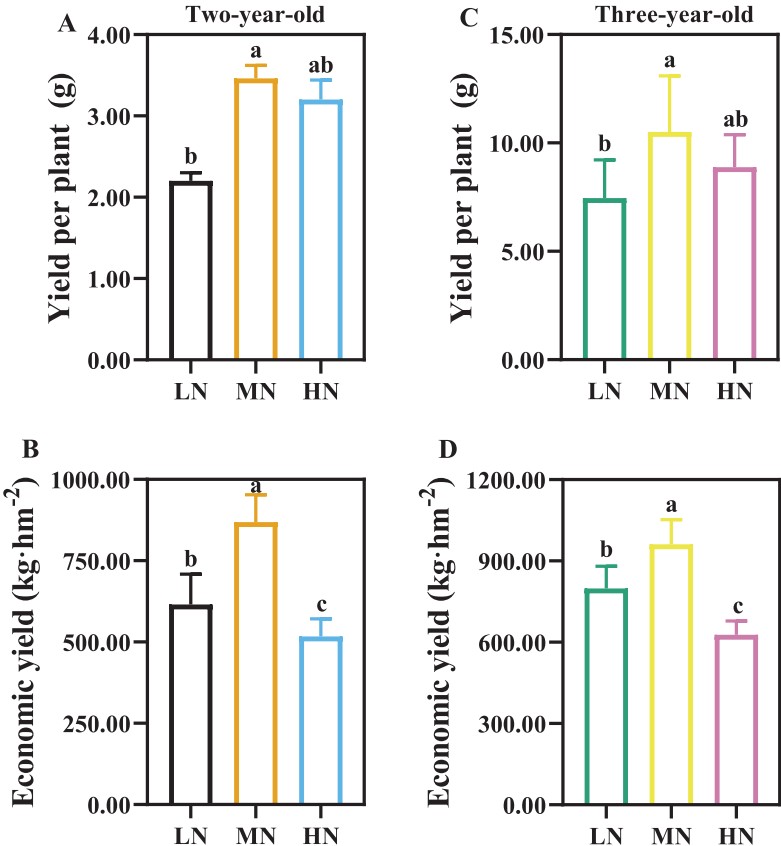

**Figure 3** **The yield of per plant (A, C) and economic yield (B, D) in two- (A, B) and three-year-old (C, D)** *Panax notoginseng* **grown under different nitrogen fertilization.** Data are mean ± SD of five independent biological replicates performed in quintuplicate (*n* = 5). Significant differences are indicated by letters (one-way ANOVA; *P* < 0.05).

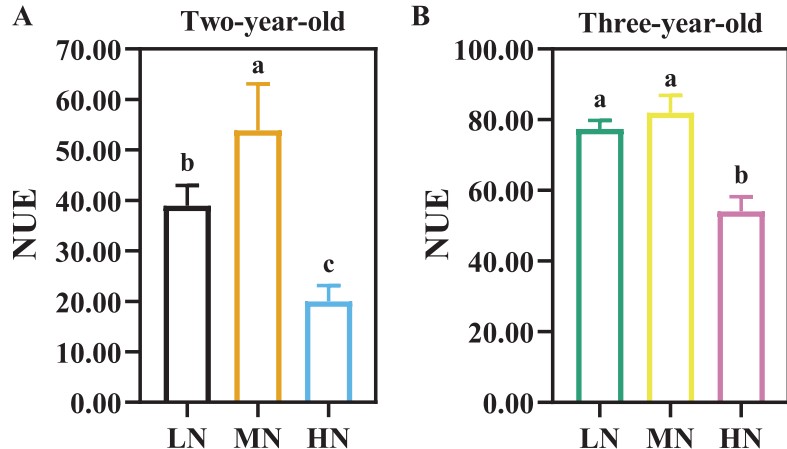

**Figure 4** **Nitrogen use efficiency (NUE) in two- (A) and three-year-old (B)** *Panax notoginseng* **grown under different nitrogen fertilization.** Data are mean ± SD of seven independent biological replicates performed in septuplicate (*n* = 7). Significant differences are indicated by letters (one-way ANOVA; *P* < 0.05).

**Table 3 Nitrogen uptake and use efficiency in *Panax notoginseng* grown under different nitrogen fertilization.**

| Variables | Two-year-old | | | Three-year-old | | |
|---|---|---|---|---|---|---|
| | LN | MN | HN | LN | MN | HN |
| NAE (kg·kg$^{-1}$) | — | 19.07 ± 4.16 a | −6.78 ± 0.65 b | — | 9.74 ± 2.07 a | −6.05 ± 0.75 b |
| NUPE (kg·kg$^{-1}$) | — | 27.98 ± 0.20 a | 14.96 ± 0.29 b | — | 17.04 ± 0.26 a | 11.69 ± 0.12 b |
| RNF (%) | — | 3.64 ± 0.20 a | 2.79 ± 0.14 a | — | 3.99 ± 0.26 b | 5.17 ± 0.06 a |
| NCR (%) | — | 34.84 ± 4.34 a | −61.41 ± 10.91 b | — | 12.91 ± 1.94 a | −22.59 ± 4.31 b |
| NPFP (kg·kg$^{-1}$) | — | 1.95 ± 0.04 a | 0.39 ± 0.08 b | — | 2.69 ± 0.02 a | 0.96 ± 0.01 b |

**Note:**
Data are mean ± SD of five independent biological replicates performed in quintuplicate ($n = 5$). Different lowercase letters among nitrogen regimes indicate significant difference (one-way ANOVA, $P < 0.05$). NAE, nitrogen agronomic efficiency; NUPE, nitrogen uptake efficiency; RNF, recovery of nitrogen fertilizer; NCR, nitrogen contribution rate; NPFP, nitrogen partial factor productivity.

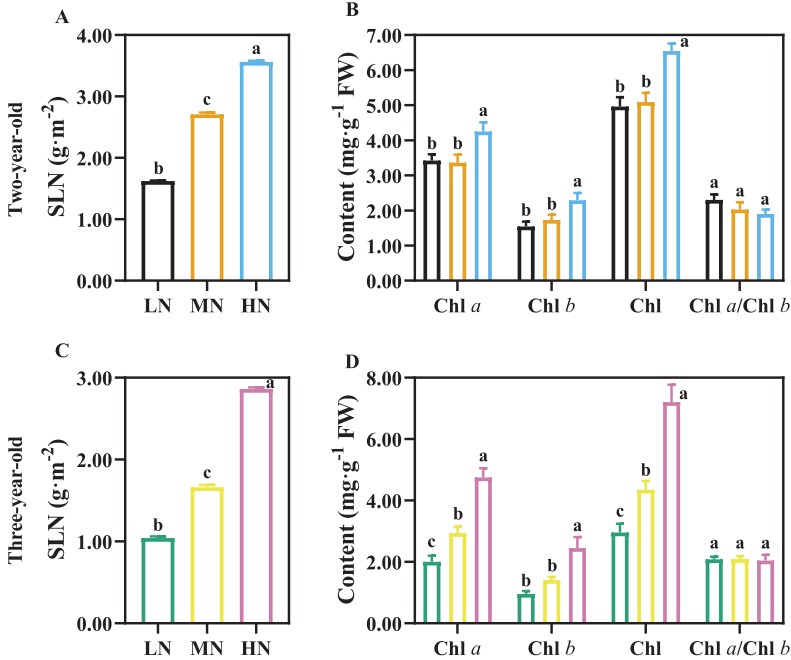

**Figure 5 The content of leaf nitrogen and chlorophyll in two- (A, B) and three-year-old (C, D) *Panax notoginseng* grown under different nitrogen fertilization.** Specific leaf nitrogen (SLN), total chlorophyll content (Chl), chlorophyll *a* content (Chl *a*), chlorophyll *b* content (Chl *b*). Data are mean ± SD of five independent biological replicates performed in quintuplicate ($n = 5$). Significant differences are indicated by lowercase letters (one-way ANOVA; $P < 0.05$).

MN conditions (Fig. 5B). Chl content increased with an increase in N application (Figs. 5B, 5D; $P < 0.05$).

$P_n$ was reduced in the LN- and HN-grown plants (Table 4, $P < 0.05$). CE (carboxylation efficiency) and $J_{max}$ were highest in two-year-old plants under MN condition. CE and $J_{max}$ were increased by 57.14% and 57.58% in MN-grown plants comparted with HN-grown *P. notoginseng*, respectively (Table 4). All variables, except the $V_{cmax}$ and $\Gamma^*$ (carbon dioxide compensation point) variables, were not significantly different in two-year-old

**Table 4 The photosynthetic-related traits in *Panax notoginseng* grown under different nitrogen fertilization.**

| Variables | Two-year-old | | | Three-year-old | | |
|---|---|---|---|---|---|---|
| | LN | MN | HN | LN | MN | HN |
| $P_n$ ($\mu mol \cdot CO_2 \cdot m^{-2} \cdot s^{-1}$) | 2.56 ± 0.16 b | 2.97 ± 0.09 a | 2.23 ±0.04 b | 0.79 ± 0.07 c | 3.01 ± 0.16 a | 1.89 ± 0.10 b |
| $P_{max}$ ($\mu mol \cdot CO_2 \cdot m^{-2} \cdot s^{-1}$) | 2.42 ± 0.27 a | 2.76 ± 0.24 a | 2.23 ± 0.29 a | 0.99 ± 0.13 c | 2.74 ± 0.20 a | 1.86 ± 0.18 b |
| LCP ($\mu mol \cdot m^{-2} \cdot s^{-1}$) | 5.95 ± 1.57 a | 4.52 ± 2.30 a | 5.18 ± 0.75 a | 28.73 ± 7.66 a | 2.83 ± 1.61 b | 10.97 ± 3.36 b |
| LSP ($\mu mol \cdot m^{-2} \cdot s^{-1}$) | 188.48 ± 20.50 a | 172.77 ± 16.53 a | 176.85 ± 19.3 a | 126.68 ± 25.04 a | 85.96 ± 6.78 b | 93.58 ± 12.08 ab |
| $R_d$ ($\mu mol \cdot m^{-2} \cdot s^{-1}$) | −0.41 ± 0.07 a | −0.46 ± 0.08 a | −0.27 ± 0.006 a | −0.95 ± 0.20 b | −0.30 ± 0.13 a | −0.70 ± 0.19 ab |
| CE ($mol \cdot mol^{-1}$) | 0.028 ± 0.001 ab | 0.033 ± 0.004 a | 0.021 ± 0.004 b | 0.004 ± 0.0001 b | 0.028 ± 0.0014 a | 0.0091 ± 0.00004 b |
| $\Gamma^*$ ($\mu mol \cdot mol^{-1}$) | 168.48 ± 5.78 b | 183.56 ± 5.28 ab | 193.10 ± 13.95 a | 145.97 ± 14.19 b | 235.42 ± 8.23 a | 141.23 ± 19.69 b |
| $J_{max}$ ($\mu mol \cdot mol^{-1}$) | 27.66 ± 0.80 ab | 31.58 ± 3.20 a | 20.04 ± 3.23 b | 29.20 ± 5.78 c | 82.76 ± 9.01 a | 57.40 ± 11.63 b |
| $V_{cmax}$ ($\mu mol \cdot mol^{-1}$) | 28.57 ± 1.32 a | 27.45 ± 1.89 a | 22.86 ± 1.61 b | 5.12 ± 0.32 b | 26.32 ± 1.89 a | 26.86 ± 1.69 a |
| $J_{max}/V_{cmax}$ | 4.99 ± 0.39 a | 5.50 ± 0.36 a | 6.71 ± 1.51 a | 4.41 ± 0.47 a | 3.25 ± 0.66 b | 2.84 ± 0.78 c |

**Note:**
Data are mean ± SD of five independent biological replicates performed in quintuplicate ($n = 5$). Different letters among nitrogen regimes indicate significant difference (one-way ANOVA, $P < 0.05$). $P_n$, net photosynthetic rate under saturated light; $P_{max}$, maximum net photosynthetic rate; LCP, light compensation point; LSP, light saturation point; $R_d$, dark respiration rate; CE, carboxylation efficiency; $\Gamma^*$, carbon dioxide compensation point; $J_{max}$, maximum electron transfer rate; $V_{cmax}$, maximum carboxylation efficiency; SLN, specific leaf nitrogen. Chl *a*, chlorophyll *a*; Chl *b*, chlorophyll *b*.

plants grown under LN and HN conditions (Table 4). $P_{max}$, LCP (light compensation point), LSP (light saturating point), $V_{cmax}$, and $R_d$ (dark respiration rate) were significantly declined in three-year-old plants under LN condition (Table 4, $P < 0.05$). For three-year-old plants, the maximum values of $P_{max}$, CE, $\Gamma^*$, $J_{max}$, $V_{cmax}$, and $J_{max}/V_{cmax}$ were obtained in MN plants (Table 4).

PNUE was declined by 26.49% and 60.65% in two- and three-year-old *P. notoginseng* grown under HN compared with MN condition, respectively (Figs. 6A, 6C; $P < 0.05$). HN induces the increase in $N_L$, and $N_C$ was reduced by 13.79% in two-year-old plants grown under HN compared with MN (Fig. 6B). $N_B$ and $N_L$ increased with the increase of N supply in three-year-old *P. notoginseng* (Fig. 6D).

### Analysis of saponin in *P. notoginseng* root

For two-year-old *P. notoginseng*, total saponins contents (%) were not significantly different N regimes (Fig. 7A). MN and HN-grown plants show 46.09% and 41.56% greater saponins yield of per plant than the LN ones (Fig. 7B, $P < 0.05$). The minimum value of total saponins (%) were recorded in three-year-old plants grown under HN condition (Fig. 7D, $P < 0.05$). For three-year-old plants, the LN and HN-grown *P. notoginseng* showed 32.58% and 28.68% lower saponins yield of per plant than the MN ones (Fig. 7E). Analogous changes in saponins yield per area and plant were recorded in two- and three-year-old plants (Figs. 7C, 7F).

### Pearson correlation analysis of parameters

Pearson correlation coefficients of 27 parameters were evaluated in *P. notoginseng* grown under different nitrogen regimes (Fig. 8). As shown in Fig. 8, root biomass was close negatively correlated with the leaf N content ($r = −0.93$) and SLN ($r = −0.87$). Root biomass was close positively correlated with N use efficiency (as reflected by NCR ($r = 0.79$), NPFP

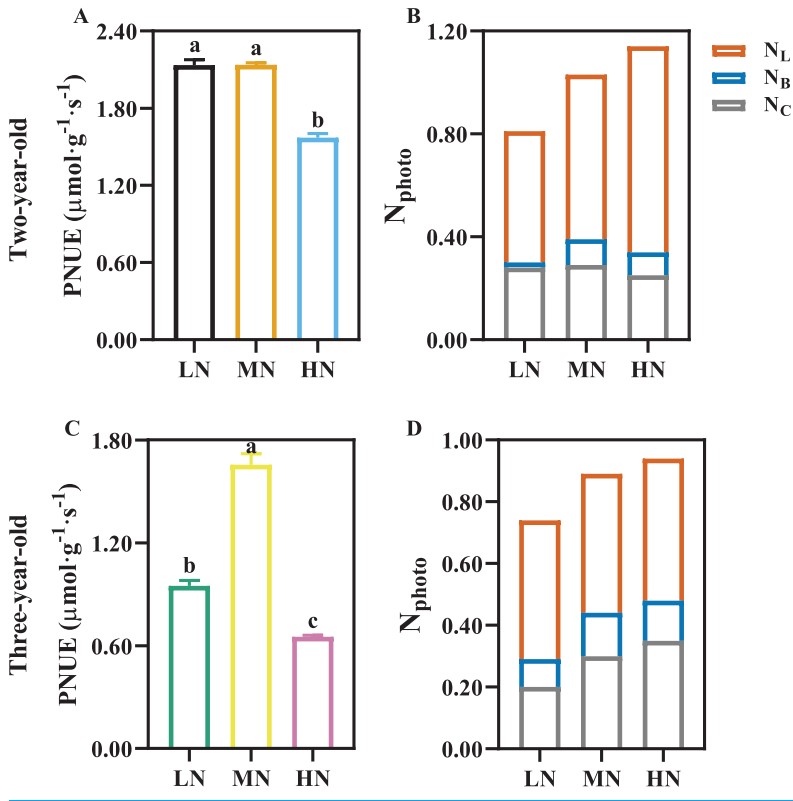

**Figure 6 The photosynthetic nitrogen use efficiency (A, C) and photosynthetic nitrogen allocation (B, D) in two- (A, B) and three-old-year (C, D)** *Panax notoginseng* **grown under different nitrogen fertilization.** PNUE, photosynthetic nitrogen use efficiency; $N_{photo}$, N content in photosynthetic apparatus; $N_L$, N content in light harvesting component; $N_B$, N content in bioenergetics component; $N_C$, N content in carboxylation component. Data are mean ± SD of five independent biological replicates performed in quintuplicate ($n = 5$). Significant differences are indicated by lowercase letters (one-way ANOVA; $P < 0.05$).

($r = 0.91$) and NUE ($r = 1.00$)). There was little correlation between root biomass and RNF, plant height and leaf area. Stem and leaf biomass were close negatively correlated with SPAD and PNUE. N application was close negatively correlated with NAE ($r = −0.87$), NCR ($r = −0.85$), NPFP ($r = −0.91$) and NUPE ($r = −0.75$). NUPE was positively correlated with the root length ($r = 0.66$) and root tuber diameter ($r = 0.59$). In addition, $P_n$ was negatively correlated with N application ($r = −0.88$), leaf area ($r = −0.57$), Chl contents ($r = −0.75$), SLN ($r = −0.44$) and leaf N content ($r = −0.45$). The relationship between saponins and $P_n$ ($r = 0.45$), root biomass ($r = 0.54$) as well as NPFP ($r = 0.57$) were positive correlation in *P. notoginseng*. Saponins content was negatively correlated with Chl content ($r = −0.67$) and SLN ($r = −0.64$).

## Comparison of the sensitivity of the different parameters in response to N regimes

A total of 20 parameters were used for three-dimensional principal component analysis (PCA). The cumulative contribution of PC1, PC2 and PC3 reached 84.80% (Fig. 9, Table S1). Thus, these three principal components could effectively explain the change of

Peerj

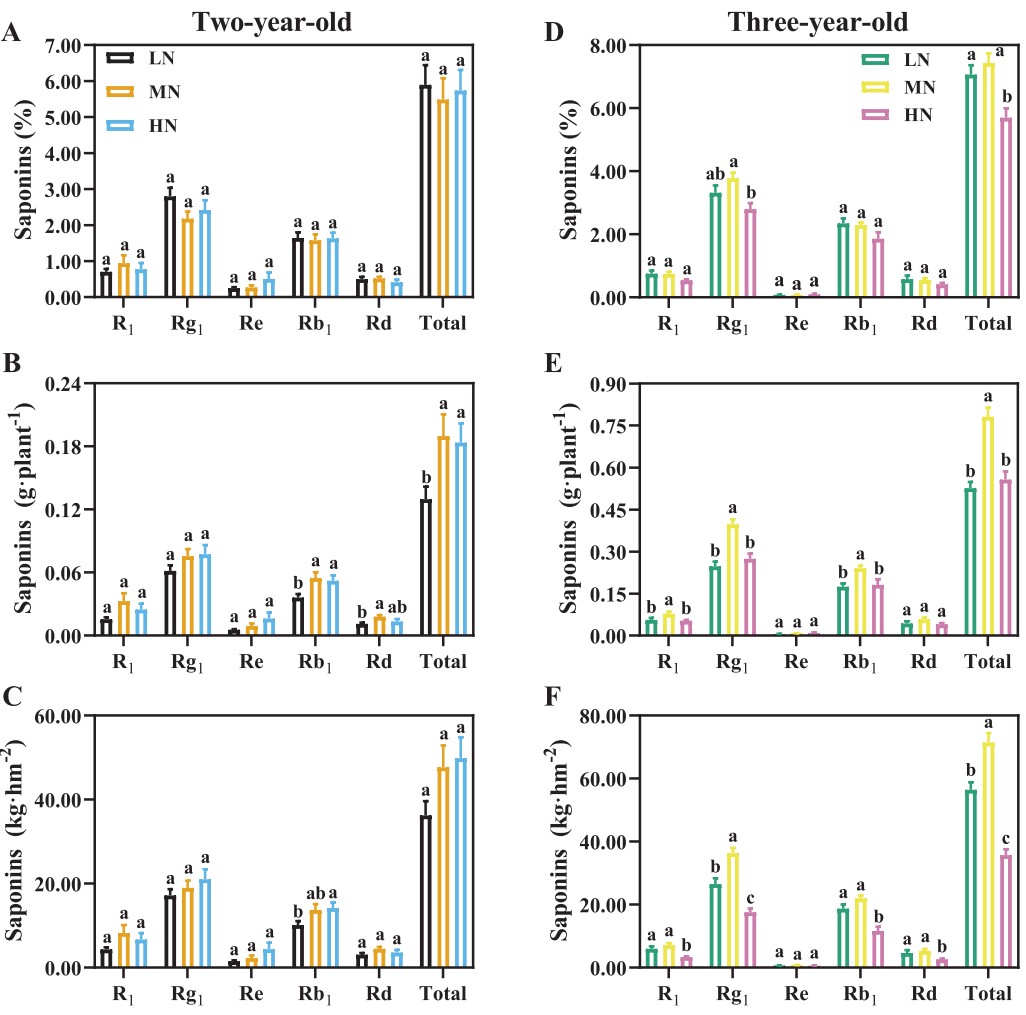

**Figure 7 Saponins content of *Panax notoginseng* root grown nitrogen regimes (A, D). The saponins yield per plant grown under nitrogen regimes (B, E). Yield of saponins per area (C, F).** Saponin type: notoginsenoside R₁, ginsenoside Rd, ginsenoside Rg₁, ginsenoside Re, and ginsenoside Rb₁. Total saponin is the sum of R₁, Rd, Rg₁, Re, and Rb₁. Data are mean ± SD of seven independent biological replicates performed in septuplicate (*n* = 7). Different lowercase letters among nitrogen regimes indicate significant difference (one-way ANOVA, *P* < 0.05).

*P. notoginseng* biomass or saponins. In PC1, the weighting coefficients of biomass parameters (as reflected by root biomass, stem biomass and leaf biomass), yield, NUE, SLN, SPAD and stem N content were larger (Fig. 9). NUE, yield, and biomass have positive correlation with PC1 and contributed more to PC1 (Fig. 9, Table S1). In PC2, the weighting coefficients of Chl parameters, root N contents, leaf area and PNUE were larger. PNUE have negative correlation with PC2 (Fig. 9, Table S1). In PC3, the weighting coefficients of $P_n$, root biomass, PNUE and yield were larger (Fig. 9, Table S1). LCP and LSP have a negative correlation with PC3 (Fig. 9, Table S1).

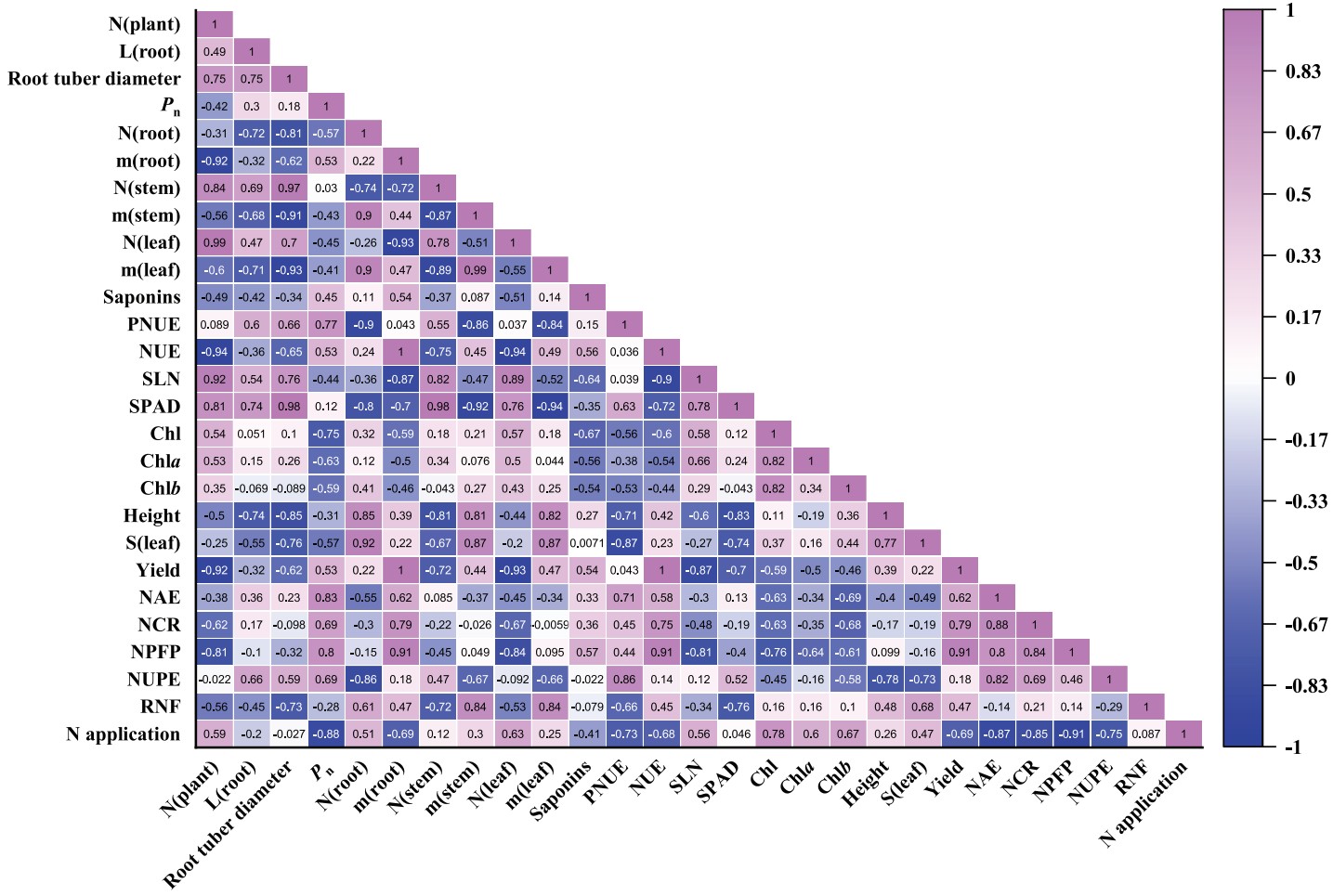

**Figure 8 Pearson correlation coefficients heatmap of all parameters evaluated in *Panax notoginseng* grown under nitrogen regimes.** Pearson correlation coefficients of 27 parameters of *P. notoginseng* under different nitrogen levels. Mediumorchid indicates positive correlation; blue indicates negative correlation. The value in each box represents the correlation coefficient. N (plant), plant total nitrogen content; L (root), root length; $P_n$, net photosynthetic rate under saturated light; SLN, specific leaf nitrogen; Chl, chlorophyll; Chl *a*, chlorophyll *a*; Chl *b*, chlorophyll *b*; N (root), N content in root; m (root), root biomass; N (stem), N content in stem; m (stem), stem biomass; N (leaf), N content in leaf; m (leaf), leaf biomass; S (leaf), leaf area; PNUE, photosynthetic N use efficiency; NUE, N use efficiency; LCP, light compensation point; LSP, light saturation point; NAE, N agronomic efficiency; NUPE, N uptake efficiency (NUPE); RNF, recovery of N fertilizer; NCR, N contribution rate; NPFP, N partial factor productivity.                

# DISCUSSION

## A "survival strategy" of inhibiting root growth under N excess

The root is a primary organ for nutrient and water absorption from the soil (*Oldroyd & Leyser, 2020*; *Sun et al., 2020*). Plants can improve N uptake by modulating root growth and architecture (*Kiba & Krapp, 2016*; *Pélissier, Motte & Beeckman, 2021*). N uptake capacity is considerably improved in N-deficient *Arabidopsis thaliana* L. by increasing the length of total root and fibrous root (*Giehl & von Wirén, 2014*). LN enhance the number, length, volume, and biomass of root to improve N uptake/use efficiency (Table 1; Figs. 2B, 4B). This fact is also verified by the positive correlation between NPUE and root length ($r = 0.66$) and root tuber diameter ($r = 0.59$, Fig. 8). LN promotes root growth, and

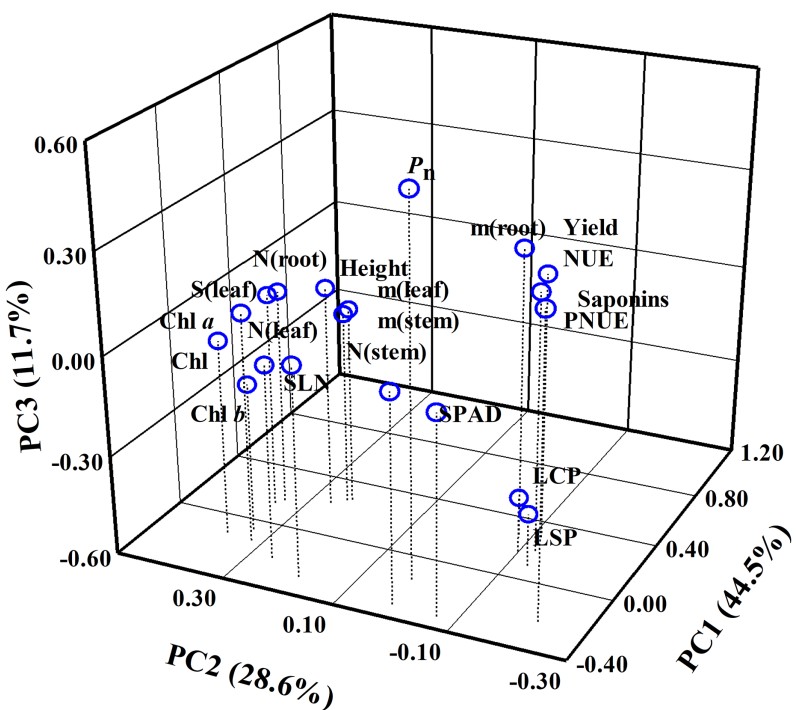

**Figure 9 Principal component analysis (PCA) using all parameters evaluated in *P. notoginseng* grown under nitrogen regimes.** Principal component analysis of 20 parameters of *P. notoginseng* under different nitrogen levels. $P_n$, net photosynthetic rate under saturated light; SLN, specific leaf nitrogen; Chl, chlorophyll; Chl *a*, chlorophyll *a*; Chl *b*, chlorophyll *b*; N (root), N content in root; m (root), root biomass; N (stem), N content in stem; m (stem), stem biomass; N (leaf), N content in leaf; m (leaf), leaf biomass; S (leaf), leaf area; PNUE, photosynthetic N use efficiency; NUE, N use efficiency; LCP, light compensation point; LSP, light saturation point.

*P. notoginseng* root adopts an "active-foraging strategy" of promoting N uptake under LN condition (*Giehl & von Wirén, 2014)*. Correspondingly, excessive N suppress length and surface area of root, and consequently inhibit root growth and biomass (*e.g.*, *Gossypium hirsutum* L., *A. thaliana* and *Cynara cardunculus* L. (*Linkohr et al., 2002*; *Leskovar & Othman, 2016*; *Chen et al., 2020*)) Previous studies are consistent with the present results that root length, NUE, RSR and RMF are inhibited by HN supply in three-year-old *P. notoginseng* (Table 1; Figs. 2, 4). HN inhibits root growth and thus reduce N uptake as observed in *Oryza sativa* L. (*Mochizuki et al., 2014*). Overall, *P. notoginseng* adopt an "active-foraging strategy" under LN condition and a "survival strategy" of inhibiting root growth under HN condition.

## N-mediated NUE and PNUE to alter biomass allocation

N uptake and use are vital for shoot and under-ground biomass allocation (*Hirose, 2011*; *Poorter et al., 2012*), and the biomass allocation is an important strategy for plants to respond to N stress (*Chen et al., 2021*). N application facilitates the leaf biomass and N content (*e.g.*, *Dodonaea viscosa* (L.) Jacq., *Lolium perenne* L. and *Betula spp.* (*Niinemets, Portsmuth & Truus, 2002*; *Wang et al., 2015*, *2020*)). Stem and leaf N content increased with an increase in N application (Fig. 1), and shoot biomass was closely correlated with N

content (Fig. 8). These results imply that more N storage of stem and leaf result in the increase of LMF and SMF when three-year-old *P. notoginseng* are exposed to HN condition (*Qiu et al., 2019*; Figs. 1, 2; Table 2). It has been reported that root N content in cotton is significantly correlated with shoot biomass (above-ground biomass) (*Wang et al., 2022*). However, root biomass was close negatively correlated with the leaf ($r = -0.93$) and stem ($r = -0.72$) N content (Fig. 8). It is a priority for *P. notoginseng* to allocate more biomass into shoot at the expense of root biomass under HN condition, and this is consistent with the results recorded by *Mehdi et al. (2018)* that RSR is decreased in *Cnicus benedictus* L. grown under N-excess condition.

Many studies have shown that N use is not positively related to N uptake (*Dong et al., 2008*; *Cheng et al., 2010*). N content is increased and NUE is declined in N-excess *Molinia caerulea* (L.) Moench (*Aerts, 1990*). This is consistent with the present results that excessive N supply could improve *P. notoginseng* N content, but reduce N use (Figs. 1, 4). However, root biomass was close positively correlated with NUE ($r = -1$), and lower NAE and NCR were obtained in HN-grown *P. notoginseng* (Fig. 8, Table 3). It might be speculated that HN-grown plants inhibit the accumulation of root biomass by reducing N use.

PNUE is one of the characteristics of physiological N use efficiency for plants, and the increased PNUE could enhance NUE and crop yield (*Ghannoum et al., 2005*; *Liu et al., 2018*). Light harvesting capacity increased with an increase in Chl and SLN (*Lei et al., 2021*). The minimum value of SLN, Chl, $N_L$ and $P_n$ were recorded in LN-grown plants (Figs. 5, 6, Table 4). The reduced light harvesting might lead to the reduced PNUE and photosynthetic efficiency in LN-grown plants (*Hikosaka, 2004*; Figs. 5, 6, Table 4). Meanwhile, CE, $J_{max}/V_{cmax}$, PNUE, $N_C$ and $P_n$ were declined in HN-grown *P. notoginseng* (Table 4, Fig. 6). These results indirectly support the view and the fact as suggested by *Zhang, Cun & Chen (2020)* and *Cun et al. (2021)* that more N exists in the form of storage proteins as N source, and thus lower Rubisco activity and C assimilation rate have been obtained in HN-grown *P. notoginseng*. Lower SLN and Chl content might lead to a decline in PNUE in LN-grown plants, and HN-induced inhibit in PNUE might be mainly due to the limitation on carboxylation efficiency. Nevertheless, N-mediated PNUE affects the accumulation of biomass under N stress (*Tofanello et al., 2021*). PNUE was close negatively correlated with the leaf ($r = -0.84$) and stem ($r = -0.86$) biomass (Fig. 8). HN-induced increase in shoot biomass might be related to the limitation to PNUE. In other words, lower PNUE was indirectly responsible for the reduction of *P. notoginseng* root biomass under HN. $P_n$ and $P_{max}$ were reduced in *P. notoginseng* under HN and LN conditions, and root biomass ($r = 0.53$) and yield ($r = 0.53$) were positively correlated with $P_n$ (Figs. 6, 8; Table 4). Our results are consistent with the previous studies that the decrease in photosynthetic efficiency under N stress inhibit the yield and biomass (*De Ávila Silva et al., 2019*). Hence, HN-induced decrease in root biomass might be derived from the suppression on photosynthetic capacity and PNUE.

### The decline in saponins contents is related to the ratio of C/N under N excess

Non-optimal nitrogen supply induced C/N imbalances, and thus affects the accumulation of secondary metabolites (*Royer et al., 2013*). C availability mainly affected by photosynthesis (*Muller et al., 2011*). Lower photosynthetic capacity causes the decline in C metabolism and the C/N, and thereby inhibits the accumulation of total phenolics, flavonoids, anthocyanins and ascorbic (C-containing metabolites) in *Labisia pumila* (Blume) Fern.-Vill. grown under high N condition (*Ibrahim et al., 2011*). It has been reported that the content of terpene decreased with increasing N addition in *Chrysanthemum boreale* M. (*Lee et al., 2005*). Total saponins content (C-containing metabolites) and $P_n$ were reduced in three-year-old *P. notoginseng* grown under HN condition (Fig. 7D, Table 4), and the $P_n$ was positively correlated with the saponins content ($r = 0.45$, Fig. 8). Lower photosynthetic capacity decreases the C/N and consequently result in a decrease of saponins content under N-excess condition (Table 4, Fig. 7), and our results are consistent with the carbon-nutrient balance hypothesis (CNB) that N excess would depress accumulation of C-containing metabolites (*Fajer, Bowers & Bazzaz, 1992*). Meanwhile, N content and N availability alters the accumulation of secondary metabolites *via* the internal C/N balance in plants (*Ibrahim & Jaafar, 2011*; *Royer et al., 2013*). HN inhibits saponins accumulation in *P. notoginseng* (Fig. 7D), and plants N content was negatively correlated with the saponins content ($r = -0.49$, Fig. 8). This is consistent with the results reported by *Chen (2005)* that the higher N content lead to lower C/N ratio, which reduces the accumulation of total phenols (C-containing metabolites) in *N. tabacum* grown under N excess condition. On the other hand, it has been recorded that phenols and flavonoids contents in *Triticum aestivum* L. are significantly declined with the decrease of NUE under N excess conditions (*Ahanger et al., 2019*). The content of saponins was positively correlated with NUE ($r = -0.56$, Fig. 8) and the minimum value of NUE was recorded in HN-grown plants (Fig. 4). This is consistent with the results that higher N content lead to lower C/N, which reduces triterpenoid (C-containing metabolites) accumulation in N-excess *Cyclocarya paliurus* (Batalin) Iljinsk. (*Qin, 2022*). The reduction in C/N caused by the lower NUE might result in the reduced accumulation of saponins under HN condition. Additionally, it has been reported that N deficiency promotes the accumulation of C-containing secondary metabolites such as phenolic (*Zhou et al., 2021*) and saponins (*Ou et al., 2020*). The present study implies that the significant difference in saponin content between LN and MN plants was not due to an imbalance in C/N (Figs. 7A, 7D). HN-induced decrease in the accumulation of saponins might be closely related to the decline in C/N.

### Moderate N application could improve the economic yield

Secondary metabolites are the quality indexes of medicinal plants, the trade-off between yield and quality should be considered in the N supply (*Ge et al., 2021*). The main root of *P. notoginseng* (Notoginseng Radix) is generally used as a traditional Chinese medicine (*Pharmacopoeia of People's Republic of China, 2020*). The economic yield of *P. notoginseng* is usually defined as the main root biomass of per unit area (*Zhang et al., 2020*). In the

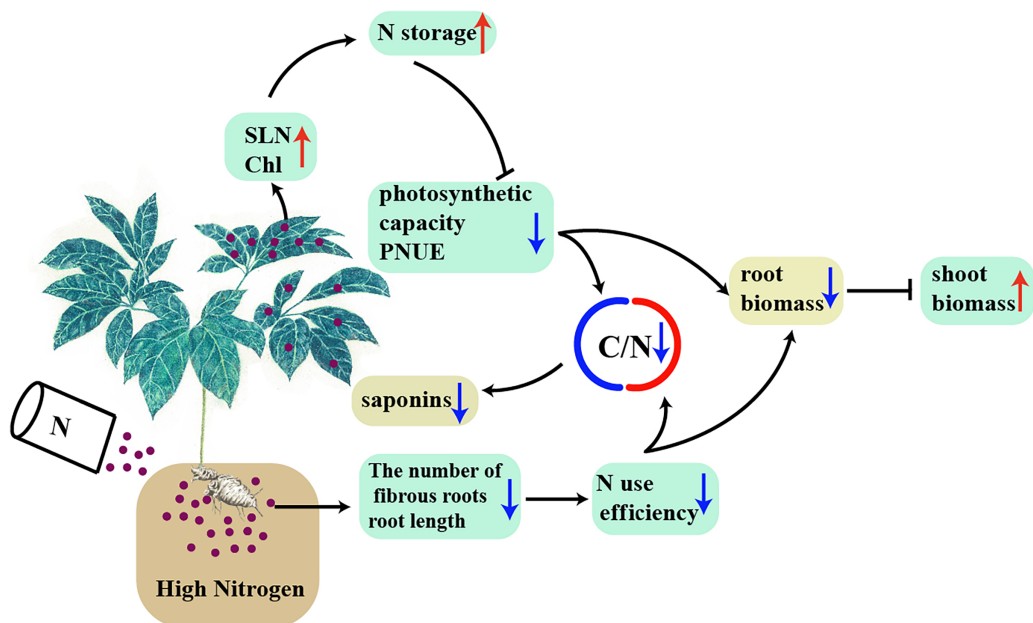

**Figure 10 A model was proposed to explain the interaction between high N and the accumulation of biomass and C-containing secondary metabolites in a N-sensitive medicinal species, such as *P. notoginseng*.** The root of N-sensitive medicinal plants adopts a "survival strategy" of inhibiting root growth under N excess, and more biomass is allocated into above-ground at the expense of root biomass by inhibiting photosynthetic capacity and N use efficiency. The reduction in C/N ratio caused by the lower N use efficiency and photosynthetic capacity result in a suppressed accumulation of saponins (C-containing metabolites) under N excess. Arrows and blunted lines designate positive and inhibitory interactions, respectively. The blue and red arrows indicate down- and up-regulation.

present results, the minimum value of economic yield (627.09 kg·hm$^{-2}$) is recorded in three-year-old *P. notoginseng* grown HN condition (Fig. 3). It implies that HN-induced decrease in economic yield may be related to the increase in incidence of root rot (*Zhang, Cun & Chen, 2020*). Additionally, total content of notoginsenoside R$_1$ and ginsenosides Rg$_1$ and Rb$_1$ has been stipulated to be not less than 5% for Notoginseng Radix in *Pharmacopoeia of People's Republic of China (2020)*. Total content of R$_1$, Rg$_1$ and Rb$_1$ were 5.19–6.80% in three-year-old roots, being higher than the standard of 5% (Fig. 7). Meanwhile, the present study found that HN advances the root biomass of per plant compared with LN, but reduces the accumulation of saponins (Figs. 3, 7E, 7F), and the lowest saponin yield of per unit area (35.71 kg·hm$^{-2}$) was recorded in HN-grown plants (Fig. 7F). This is consistent with the results that N application could enhance biomass, but reduce saponins accumulation in N-excess plants (e.g., *Centella asiatica* L. and *Stevia rebaudiana* (Bertoni) Hemsl. (*Müller et al., 2013*; *Barbet-Massin et al., 2015*)). However, despite higher RSR and RMF, LN-grown plants generally have a lower yield of saponin (Figs. 2, 7B, 7C, 7E, 7F). This might be related to the fact that the main root of *P. notoginseng* has been commonly used for estimating crop yield and medicinal quality (*Pharmacopoeia of People's Republic of China, 2020*). In short, saponins yield and economic yield are reduced in HN-grown *P. notoginseng*.

## CONCLUSION

A model was proposed to explain the interaction between high N and the accumulation of biomass and C-containing secondary metabolites in a N-sensitive medicinal species, such as *P. notoginseng* (Fig. 10). In conclusion, the root of N-sensitive medicinal plants adopts a "survival strategy" of inhibiting root growth under N excess, and more biomass is allocated into above-ground at the expense of root biomass by inhibiting photosynthetic capacity and N use efficiency. The reduction in C/N ratio caused by the lower N use efficiency and photosynthetic capacity result in a suppressed accumulation of saponins (C-containing metabolites) under N excess. Overall, N excess reduce the yield of root and of C-containing secondary metabolites in an N-sensitive medicinal species such as *P. notoginseng*.

### Funding

This research was supported by the National Natural Science Foundation of China (32160248 and 81860676), the Major Special Science and Technology Project of Yunnan Province (202102AA310048), the National Key Research and Development Plan of China (2021YFD1601003), and the Innovative Research Team of Science and Technology in Yunnan Province (202105AE160016). The funders had no role in study design, data collection and analysis, decision to publish, or preparation of the manuscript.

### Grant Disclosures

The following grant information was disclosed by the authors:
National Natural Science Foundation of China: 32160248 and 81860676.
Major Special Science and Technology Project of Yunnan Province: 202102AA310048.
National Key Research and Development Plan of China: 2021YFD1601003.
Innovative Research Team of Science and Technology in Yunnan Province: 202105AE160016.

### Competing Interests

The authors declare that they have no competing interests.

### Author Contributions

- Zhu Cun conceived and designed the experiments, performed the experiments, analyzed the data, prepared figures and/or tables, authored or reviewed drafts of the article, and approved the final draft.
- Hong-Min Wu conceived and designed the experiments, performed the experiments, analyzed the data, prepared figures and/or tables, authored or reviewed drafts of the article, and approved the final draft.
- Jin-Yan Zhang conceived and designed the experiments, performed the experiments, analyzed the data, prepared figures and/or tables, authored or reviewed drafts of the article, and approved the final draft.
- Sheng-Pu Shuang conceived and designed the experiments, performed the experiments, analyzed the data, prepared figures and/or tables, authored or reviewed drafts of the article, and approved the final draft.
- Jie Hong conceived and designed the experiments, performed the experiments, analyzed the data, prepared figures and/or tables, authored or reviewed drafts of the article, and approved the final draft.
- Tong-Xin An conceived and designed the experiments, performed the experiments, analyzed the data, prepared figures and/or tables, authored or reviewed drafts of the article, and approved the final draft.
- Jun-Wen Chen conceived and designed the experiments, performed the experiments, analyzed the data, prepared figures and/or tables, authored or reviewed drafts of the article, and approved the final draft.

## Data Availability

All raw data are available in the Supplemental File.

## Supplemental Information

Supplemental information for this article can be found online at http://dx.doi.org/10.7717/peerj.14933#supplemental-information.

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
