# Peer review of "High nitrogen inhibits biomass and saponins accumulation in a medicinal plant Panax notoginseng"

_PeerJ, doi:10.7717/peerj.14933_

## Round 0.1 · original submission · Major Revisions

The manuscript requires major revisions according to the comments of both reviewers, thanks

Reviewer 1 ·

Basic reporting

Dear Authors,
The presented study describes effects of nitrogen (N) fertilization on growth as well as on physiology and biochemistry of Panax notoginseng (Burkill) F. H. Chen. Overall, it was showed that excess of N reduces yield of biomass and metabolites in the studied species. It was also suggested that the studied species employ reaction aiming on reduction of negative effects of N over-supplementation.
Overall, I think that this article is interesting and there is some cool biology within it. However, there are several flaws making it in this version hard to follow or sometimes unreadable. The biggest flaw is lack of detailed description of statistical analysis (I believe that it can be analyzed using two-way ANOVA). The analysis of correlation is very important here but can must be improved (also including presentation of results). Additionally, I feel that the ‘Introduction’ and ‘Discussion’ sections are too descriptive.
Thus, I recommend to revise this manuscript (major revision). As soon as all the major issues will be clarified, the manuscript will be of acceptable quality.

Experimental design

Overall, the experimental design meets scientific standards. However, statistical analysis must be improved and described in detail in revised version of the manuscript.

Validity of the findings

Overall, thefindings are valid and meets scientific standards. However, the 'Discussion' must be partially reworked. Additionally, 'Result' section needs some improvements.

Additional comments

MAJOR ISSUES

1) This manuscript contains 20 items in total (14 Figures and 6 Tables). This is definitely too much. Additionally, there are some problems with captions. I suggest to do the following actions:
a) Principal request (mandatory): please correct y-axis values. Crossing with the x-axis should always be at the 0 value to reduce bias for the audience. Check all the figures and fix this (e.g., Figure 1, 3, 5, 8, 9).
b) Mandatory: Figures 4, 6, 9, 11, 12 and Table 4 – delete it, merge data from all of them (or conduct new analysis) and present as one (or two if u decide to separate two- and three-yo plants) table or heatmap. In all the cases, please make sure that both correlation value and significance are readable (by color, symbol, presentation of exact value – it is up to you).
c) Figure 13 – Why did you not select PCA based on correlations? In my opinion it can be easier for interpretation, especially in the context of practical implication and data from correlations that you already presented. Additionally, marks for two- and three-yo-plants are the same (please correct this).
d) Remove Figure 14 as it can be described in “Discussion” (up to decision of the Editor and the Authors)
e) Mandatory: Table 6 with PCA loadings – make it as Supplementary Material. Additionally, show PC3, as you model explained 73.5% of variance on the basis of two first components – it can be very valuable to check contribution of PC3 (even just as values of loadings; for generation of Figure 13 please remain as it is – PC1 and PC2).
f) Mandatory: Figure captions: description “Green represents low nitrogen (LN), bule represents moderate nitrogen (MN), and red represents high nitrogen (HN)” is not needed because it is already self-containing (variants are named in the figure)
g) Delete “medicinal plant” from each caption.
h) Mandatory: replace “level” in each caption with more adequate term, e.g., “nitrogen availability” or “nitrogen fertilization” but not “level”.
i) Mandatory: in all the caption always give name of statistical test used for comparisons.

2) Please always use SI units and correct formula for each units, e.g., remove all “/”; thus always use interpunct (middle dot). Please check where applicable.

3) Statistical analysis. It is documented very poorly. Please check comments on L214-217.

4) The results (L219-L305). Always provide information if the changes were significant. Additionally, always describe magnitude of change, e.g., 50% more or 1.5 times more or 150% of control variant, etc. Check where applicable in lines L219-305.

5) Please change style of writing, namely, when you provide information about published data and findings, please state that, i.e., “roots are smaller when plants are subjected to elevated availability of N (e.g., in Species1, Species2, Species3 [from references])” but not “roots of Species1, Species2 and Species3 [from references] are smaller when plants are subjected to elevated availability of N”. This is a classic example of a generalization problem. In such case, what about hypothetical Species4 - will it be similar to all the other mentioned ones, or not? How about species you studied? Please check it throughout the manuscript and fix where applicable (e.g., L51-64, L65-83, etc.).

6) Raw data:.pzfx files could be unreadable for many researchers who do not use Prism. Please convert it to more common format.

7) Raw data: each dataset (in this case sheet) needs description below data. Additionally, for each parameter, unit should be provided.

Detailed (line-by-line) list of comments
L51: Change “factor” for “nutrient”
L51-64: It can be better to present that these symptoms are typical for N deficiencies in all the higher terrestrial plants (remove names of species). Additionally, you should find review article – it is better to cite review in this case in order to reduce extensive citation.
L65-83: Similarly, please find review to cite effects of optimal and/or luxuriant N fertilization. This paragraph needs more generalization, as positive effects of N on photosynthesis was reported for numerous plant species.
L84-103: This is written well, but answer here this questions: which one yield (from N-oversupplied or N-starved plants) is better in terms of specialized metabolites and thus, industry?
L138: Provide volume of pots
L139: Very mandatory: provide full description of photothermal conditions, as they are key for studies on effects of macroelemental fertilization. Provide type of lights, PAR, photoperiod, etc.
L143-149: Provide names of N-, P- and K-containing compounds, their source and, if they were laboratory grade chemicals, their purity.
L163: Which extraction solvent was used? Please provide also wavelengths and name of device used for determination.
L210-212: Provide source of standards and their purity.
L214-217: How normality was tested, did the data meet requirements for ANOVA, how variances were tested, which ANOVA was used (one- or two-way), which post hoc was used? How correlations were checked? How PCA was conducted – which variables were active and which were additional (if there were any)?
L309-310: I do not agree. Is root morphology a reason why N uptake is altered or N uptake shapes root morphology? Or, these two traits (root morphology and N uptake) are strongly connected and work as negative loop feedback?
L312-315: This is result, not discussion.

Supplementary Figure: Are the presented plants two or three year-old?
Figure 10: Provide full name of metabolites, in Figure or in caption.

Reviewer 2 ·

Basic reporting

This manuscript is fairly well-written albeit its length can be reduced by removing excessive information in the Introduction and Discussion sections.

The number of tables and figures accompanying the text should either be reduced or multiple figures on the same aspect to be combined.

Experimental design

Standard procedures were used in this study, sufficiently described, and in most cases, appropriately referenced. Nonetheless, improvements are suggested:

1. Please provide the details of the plantlets (source, authentication) used in this study.
2. Why two- and three-years old plantlets were used?
3. How did the authors decide on the low, moderate and high concentrations of N?
4. For the HPLC analysis, please provide the details of the standards used especially the % purity.
5. Extraction of saponins need to be clarified. Is soaking of plant samples in methanol the optimal way to extract saponins?

Validity of the findings

Suggestions for improvement.

1. This study focuses on the effects of N content of plant biomass and saponin content, hence, information such as the HPLC chromatograms should be included for better visualisation and comparison.

2. The authors postulated that the reduction in saponins content could be due to the C/N ratio and excessive amount of N. This postulation can be strengthened by drawing more comparisons from related study.

3. Multiple correlations were performed with reported R values ranging from 0.2 to 0.9. However, in many cases, the terms used are unclear and inconsistent. Please distinguish between positive and negative correlations, and whether the correlations are strong, moderate or weak.

Additional comments

Overall this is an interesting study with potential insights into the mechanism of accumulation of secondary metabolites in the ginseng plant. Extensive data collected and compiled but data analysis and interpretation needs improvement. Please consider to shorten the manuscript and reorganise the figures and tables. The manuscript needs to be improved before it can be considered for publication.

---

## Round 0.2 · Minor Revisions

the manuscript minor revision suggested by the reviser:

1. The Figure S4 should be included in the manuscript to illustrate the (hypothesised) relationships between the parameters of interest such as N content, saponins and biomass.

2. In the Statistical analysis section and figure legends, clarify what is represented by mean +/- SD - do the values represent technical replicates of plants or analysis?

Reviewer 1 ·

Basic reporting

Dear Authors,
I believe that the current version of this manuscript, with slight modifications, could be accepted for publication.

Experimental design

The experimental design is correct and the manuscript meets standards of PeerJ.

Validity of the findings

The findings are now validate.

Additional comments

1. Please check once again units. I believe that mL should be converted into cm^3.
2. Figure 9: Please slightly rework this figure, i.e., please edit descriptions of axis x,y and z. I believe that work of Kołodziejek et al., 2010 (doi: 10.2478/s11756-010-0009-7) can a guide for this action.

Reviewer 2 ·

Basic reporting

The authors have addressed the issues raised by the reviewers. The manuscript's scientific quality looks good. I do not have further questions but some suggestions for consideration:

1. The Figure S4 should be included in the manuscript to illustrate the (hypothesised) relationships between the parameters of interest such as N content, saponins and biomass.
2. In the Statistical analysis section and figure legends, clarify what is represented by mean +/- SD - do the values represent technical replicates of plants or analysis?

Experimental design

No comment

Validity of the findings

No comment

---

## Round 0.3 · accepted · Accept

I am writing to inform you that your manuscript - High nitrogen inhibits biomass and saponins accumulation in a medicinal plant Panax notoginseng - has been Accepted for publication in PeerJ